

# A unique mating strategy without physical contact during fertilization in Bombay Night Frogs (*Nyctibatrachus humayuni*) with the description of a new form of amplexus and female call

Bert Willaert[1], Robin Suyesh[2], Sonali Garg[2], Varad B. Giri[3], Mark A. Bee[4] and S.D. Biju[2]

[1] Hansbeke, Belgium
[2] Systematics Lab, Department of Environmental Studies, University of Delhi, Delhi, India
[3] Research Collections, National Centre for Biological Sciences, Bangalore, Karnataka, India
[4] Department of Ecology, Evolution, and Behavior, University of Minnesota–Twin Cities Campus, St. Paul, Minnesota, USA

Corresponding authors
Bert Willaert,
bert.willaert@gmail.com
S.D. Biju,
sdbiju.du@gmail.com

## ABSTRACT

Anurans show the highest diversity in reproductive modes of all vertebrate taxa, with a variety of associated breeding behaviours. One striking feature of anuran reproduction is amplexus. During this process, in which the male clasps the female, both individuals' cloacae are juxtaposed to ensure successful external fertilization. Several types of amplexus have evolved with the diversification of anurans, and secondary loss of amplexus has been reported in a few distantly related taxa. Within *Nyctibatrachus*, a genus endemic to the Western Ghats of India, normal axillary amplexus, a complete loss of amplexus, and intermediate forms of amplexus have all been suggested to occur, but many species remain unstudied. Here, we describe the reproductive behaviour of *N. humayuni*, including a new type of amplexus. The dorsal straddle, here defined as a loose form of contact in which the male sits on the dorsum of the female prior to oviposition but without clasping her, is previously unreported for anurans. When compared to known amplexus types, it most closely resembles the form of amplexus observed in Mantellinae. Furthermore, we prove that, opposed to the situation in most anurans, male semen release happens before egg deposition. We hypothesize that the male ejaculates on the female's dorsum and that sperm subsequently runs from her back and hind legs before fertilizing the eggs. A second feature characterizing anuran breeding is the advertisement call, mostly produced solely by males. Despite recent descriptions of several new *Nyctibatrachus* species, few studies have explored their vocal repertoire. We describe both the male advertisement call and a female call for *N. humayuni*. The presence of a female call has not been reported within Nyctibatrachidae, and has been reported in less than 0.5% of anuran species. Altogether, our results highlight a striking diversity and several unique aspects of *Nyctibatrachus* breeding behaviour.

## INTRODUCTION

Anurans exhibit considerable diversity in their reproductive modes, with differences occurring, for example, in oviposition sites, larval development, and parental care (*Crump, 1974*; *Haddad & Prado, 2005*; *Crump, 2015*). One behaviour characteristic of nearly all anuran species' reproduction is amplexus. This behaviour, whereby the male takes the female in an embrace, is considered to have evolved to optimize successful fertilization of the eggs. Because fertilization is external in most anurans, the juxtaposition of the male and female cloacae during amplexus helps to synchronize egg deposition and the release of sperm (*Duellman & Trueb, 1986*: 68–70; *Wells, 2007*: 452–458). Several forms of amplexus have been described, with inguinal and axillary being the most widespread. Inguinal amplexus, in which the male clasps the female around her waist, is considered the ancestral state and is found in the oldest anuran lineages (*Duellman & Trueb, 1986*: 68–70; *Wells, 2007*: 452). Axillary amplexus, in which the male grasps the female in her armpits, is sometimes considered to be more efficient because the cloacae are better juxtaposed (*Rabb & Rabb, 1963*). While there is no evidence that axillary amplexus results in higher fertilization rates than inguinal amplexus, species with axillary amplexus may require less time to lay their eggs, thereby reducing an amplectant pair's vulnerability to predators (*Wells, 2007*: 456). Other forms of amplexus, in addition to axillary amplexus and inguinal amplexus, have evolved in some lineages (*Duellman & Trueb, 1986*: 68–70). A complete loss of amplexus has been observed in a few distantly related taxa (*Limerick, 1980*; *Kunte, 2004*; *Zhang et al., 2012*).

Another distinctive behaviour associated with anuran reproduction is calling. The vocal repertoire of male frogs and toads is well known, with the advertisement call, which is produced to attract mates and signal presence towards other males, being present in all but a few species (*Wells & Schwartz, 2007*). Male advertisement calls are species specific and can convey information about the signaller to other individuals. A key function of male advertisement calls is in premating species isolation (*Gerhardt & Huber, 2002*). It is, therefore, important to record and describe advertisement calls, as they can provide useful information in both taxonomical and evolutionary frameworks. Female calls are less well known because they are rare and seldom observed. For example, females in only a few anuran species are known to produce true advertisement calls (e.g., *Emerson, 1992*; *Bush, Dyson & Halliday, 1996*; *Bush, 1997*; *Bosch & Márquez, 2001*). The use of courtship calls by females engaged in close-range interactions with males has been reported in a number of species (e.g., *Given, 1993*; *Tobias, Viswanathan & Kelley, 1998*; *Bosch, 2002*; *Shen et al., 2008*; *Cui et al., 2010*). In some species, females produce a territorial call upon disturbance (*Capranica, 1968*; *Wells, 1980*; *Stewart & Rand, 1991*). Females in some anuran species also produce a release call when they are amplexed but unreceptive, or amplexed by an undesired male (*Brzoska, Walkowiak & Schneider, 1977*; *Gollmann, Benkö & Hödl, 2009*).

The genus *Nyctibatrachus Boulenger, 1882* is endemic to the Western Ghats of India and comprises 28 known species, many of which have only recently been described (*Das & Kunte, 2005*; *Biju et al., 2011*; *Gururaja et al., 2014*; *Frost, 2015*). *Nyctibatrachus* species

vary in snout-vent length (SVL) from 10.0 mm (*N. minimus Biju et al., 2007*) to 76.9 mm (*N. grandis Biju et al., 2011*) and are either stream-associated or found in the leaf litter, the latter making use of small puddles for reproduction (*Biju et al., 2007*; *Biju et al., 2011*; *Van Bocxlaer et al., 2012*). Information about their natural history is scarce, and the conservation status of many species is unknown, with only 15 out of the 28 species having been assessed by the IUCN Red List (including four that are considered data deficient) (*Biju et al., 2011*; *IUCN, 2014*). Furthermore, male advertisement calls have only been described in four species (*Kuramoto & Joshy, 2001*; *Gururaja et al., 2014*).

Despite their poorly understood ecology, several studies suggest the presence of interesting forms of reproductive behaviour (*Kunte, 2004*; *Biju et al., 2011*; *Gramapurohit, Gosavi & Phuge, 2011*; *Gururaja et al., 2014*). In all *Nyctibatrachus* species, egg clutches are deposited on rocks or vegetation overhanging water, and tadpoles fall in the water after hatching, where they continue their development and metamorphosis (*Biju et al., 2011*; *Gururaja et al., 2014*). Different kinds of amplexus behaviours have been described within this genus. Pairs of *N. kumbara Gururaja et al., 2014*, for example, perform a short axillary amplexus followed by a handstand. The female then deposits the eggs from this upside down position directly after the male has dismounted (*Gururaja et al., 2014*). Afterwards, males of *N. kumbara* cover the deposited eggs with a layer of mud, a behaviour previously unknown for anurans. In *N. aliciae Inger et al., 1984*, *N. humayuni Bhaduri & Kripalani, 1955*, *N. jog Biju et al., 2011* and *N. minor Inger et al., 1984* a short, loose physical contact between the male and female takes place as the male sits on the dorsum of the female but does not clasp her as in a normal axillary amplexus (*Biju et al., 2011*; *Gramapurohit, Gosavi & Phuge, 2011*). In other species, such as *N. petraeus Das & Kunte, 2005*, amplexus behaviour may be completely absent, with the female depositing the eggs prior to the male fertilizing them (*Kunte, 2004*; *Das & Kunte, 2005*). On one occasion, amplexus was observed in *N. petraeus* five minutes before egg deposition took place, and the author therefore considered this to be a type of 'pseudo-amplexus' (*Kunte, 2004*).

Together the available data indicates a wide range of breeding-associated behaviours in *Nyctibatrachus*, with a new form of amplexus and an intrageneric variation in amplexus types previously unreported in anurans. To better understand the reproductive behaviour of *Nyctibatrachus* frogs, we observed breeding and associated vocalizations in a wild population of *N. humayuni*. Here, we describe both the male advertisement call and a female call, and discuss the different behavioural steps involved in reproduction based on field observations.

## MATERIALS AND METHODS

### Field surveys and behavioral observations

We spent a total of 40 nights in the field during July and August 2010 and 2012 studying a population of *Nyctibatrachus humayuni*. The population was located in a dense forest near Humbarli village, Koyna, Satara District, Maharashtra (coordinates 17°24′10.8″N, 73°44′13.2″E, 827 m asl) (Fig. S1A). Males were easily located by their calls while females, which were harder to find, were more often encountered by chance. Sex was determined by the presence of femoral glands in males and their absence in females (*Biju et al., 2011*).

The moment a female was observed approaching a male, we started filming the event using a camera with infrared function (Sony HDR-XR 550VE). In most such instances, we used infrared light to avoid disturbing the animals and affecting their behaviour. Movies were analysed with iMovie 8.0.6 (Apple Inc.) to determine the duration of different steps of reproduction. In order to test the assumption that fertilization happens after the female has left the oviposition site (*Gramapurohit, Gosavi & Phuge, 2011*), Ziploc® bags were placed around egg clutches directly after deposition on five occasions to hinder male contact with the eggs. To determine the duration of embryo development, deposited and fertilised clutches were monitored every evening until hatching occurred.

## Call recording

The advertisement calls of eight males and the calls of one female were recorded on a solid-state digital recorder (Marantz PMD620, 44.1 kHz sampling rate, 16-bit resolution) using a directional shotgun microphone (Sennheiser ME 66). A minimum of 20 calls were recorded per individual. Microphones were handheld and positioned at a distance of approximately 75 cm from the target animal. Sounds were monitored in real time using headphones (Sony MDR-V500). At the end of each recording, the calling individual was captured and its SVL was measured to the nearest 0.1 mm using digital callipers. A portable digital balance was used to measure body mass to the nearest 0.01 g. These two measures of body size were used to compute a measure of body condition (i.e., length independent mass) following *Baker (1992)*. Condition was estimated as the residuals from a regression of the cube root of mass on SVL divided by SVL. We used these measures of SVL body mass and body condition to assess whether any call properties were correlated with body size and condition. Recorded frogs were released at their calling site immediately after obtaining body size measurements. To avoid recording the same individual twice, we only recorded animals that were widely spaced (> 15 m). Since the study area was large (ca. 250 m × 30 m) and males of this species are territorial, being found sitting next to or on previously deposited egg clutches on subsequent nights, the chance of recording the same individual multiple times was negligible (*Gramapurohit, Gosavi & Phuge, 2011*). As call properties can vary with temperatures in anurans (*Gerhardt & Huber, 2002*), we recorded both dry bulb and wet bulb air temperatures (± 0.2 °C) at the animal's calling site using a thermometer (Jennson Delux).

This study was conducted with permissions and guidelines from the responsible authorities in the State Forest Department of Maharashtra. Study permit: D-22 (8)/Research/4543/2012-13, dated 1-03-2012. This study did not sample animals for any captive or laboratory studies. All observations were made in the wild. Recorded frogs were released back at their calling site immediately after measuring the body size and body mass.

## Call analysis

We used Raven Pro 1.4 (*Charif, Waack & Strickman, 2010*) to measure 32 acoustic properties for each of the 160 advertisement calls recorded from the eight males. Raven's waveform display was used to measure 21 temporal properties, while 11 spectral

properties were measured using the spectrogram slice view (1024pt. FFT, Hanning window). A description of the measured properties is provided in Table S1A. Properties are analyzed after *Bee, Suyesh & Biju (2013a)* and *Bee, Suyesh & Biju (2013b)*. Coefficients of variation (CV = standard deviation/mean) were computed to describe patterns of both within-individual ($CV_w$) and between-individual ($CV_b$) variation in call parameters and are expressed her as percentages (*Gerhardt, 1991*). Correlation of any of the acoustic parameters with physical characteristics (SVL, body mass and body condition) and temperature was explored by performing Spearman rank correlations using the software package Statistica v7.1 (Statsoft). Due to the low sample size, these correlation analyses should be considered exploratory and are not intended to test any specific hypotheses. Five temporal and four spectral properties were measured for the 20 calls recorded from one female (Table S1B). Since we only recorded the call of a single female, descriptive statistics and correlations with other parameters could not be carried out.

## RESULTS

### Reproductive behaviour

Breeding took place in and around a stream in which the water level heavily depended on the weather conditions (Figs. S1B and S1C). Males were observed calling from different positions along the stream. They were perched on rocks, vegetation and fallen trees, all bordering or overhanging the water (Video S1). Their vertical position ranged from partially submerged to up to four metres high on trees. Females were observed moving slowly between the territories of different males. A schematic overview of the reproductive sequence is shown in Figs. 1 and 2, and videos of the different steps are provided as Supplementary Information.

When a female approaches a male (Figs. 1A and 2A; Video S2), she sits in front of him and creeps backwards until her abdomen is placed over his head, making physical contact (Figs. 1B and 2B; Video S3). At this moment, the male mounts the female. If the male does not react, the female will repeat this physical contact. When mounted, the male rests on the female without grabbing her in a firm amplexus, and instead uses his hands to hold on to the leaf or branch on which the pair sits (Figs. 1C and 2C; Video S4). Lateral movements in the male's flank were observed during this stage in a few of the breeding sequences (Video S4). This loose form of amplexus is of short duration, lasting on average 768 s (n = 21, range: 338–1670 s). At the end of this embrace, the female strongly and repeatedly arches her back followed by the male immediately dismounting (Fig. 1D; Video S5), a behaviour known from other species too (*Gosner & Rossman, 1959*). The female deposits eggs in a single bout immediately after the male dismounts (Figs. 1E and 2D; Video S6). Oviposition takes only a few seconds and occurs without any form of physical contact between the sexes. The female then remains motionless, with her hind legs stretched around the freshly deposited clutch (Figs. 1E and 2D; Video S6). It takes on average 479 s (n = 17, range: 260–961 s) before the female moves after egg laying. After this first movement, it takes another 728 s (n = 11, range: 405–1692 s) before she jumps back into the stream. During this period, the male is sitting nearby. There is, however, no physical contact between the two sexes after egg deposition.

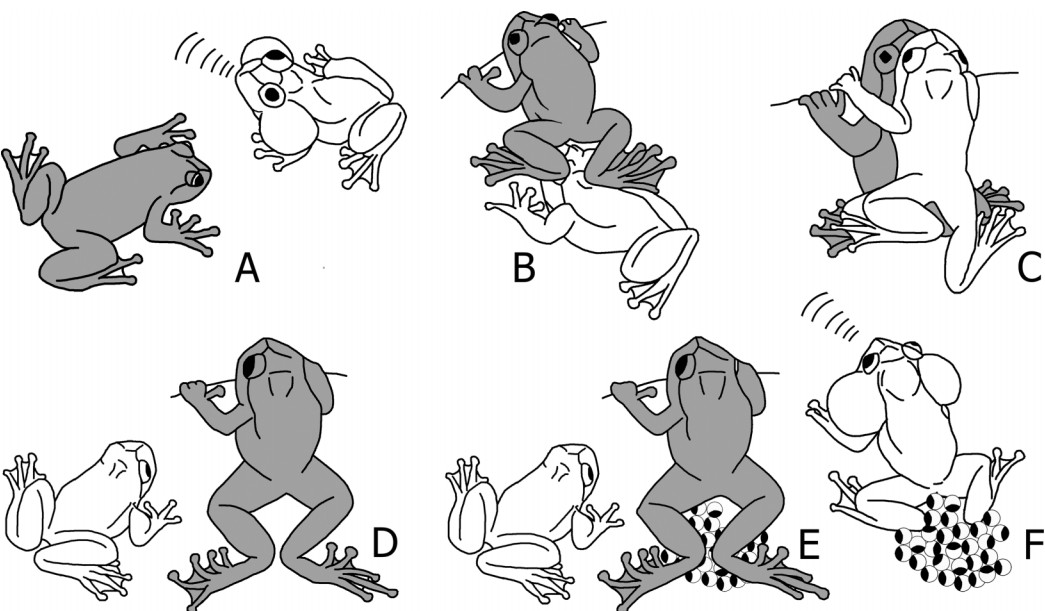

**Figure 1** **Schematic sequence of reproductive behaviour in *Nyctibatrachus humayuni*.** (A) A female approaches a calling male. (B) The female sits in front of the male and creeps backwards, until her feet touch the male's head. (C) The male mounts the female and forms a dorsal straddle, and most likely sperm is released on her back during this moment. (D) When the female is about to deposit the eggs, the male dismounts. (E) Immediately after the male dismounts, the female deposits the eggs and remains motionless with her hind legs stretched around the eggs. (F) After oviposition, the female leaves the oviposition site, and the male sits on or near the eggs and continues to call. The male is drawn white and the female grey.

During several of the observed breeding sequences, the male crawled back over the eggs after the female had moved away (Figs. 1F and 2E), but we did not see this on every occasion. This prompted us to test whether fertilization had already happened during an earlier stage. We tested this by placing a Ziploc® bag around five egg clutches directly after deposition, and in this way prevented the male from having any further contact with the eggs (Video S7). Interestingly, in all five clutches, the fertilization rate was 100%. In at least half of the observations we made (nine observations), reproduction was interrupted by one or both of the frogs falling into the water (Video S6, Examples 6 and 7). Fallen individuals returned to the same position to continue the mating sequence, except when strong currents swept them away. Males showed high site fidelity, with new eggs being deposited directly next to present clutches or in close vicinity ($\leq$ 50 cm). When a female approached a male calling from a position away from his previously fertilized clutches, the male moved towards those eggs upon the first contact while continuing to produce advertisement calls as the female followed. Males are known to defend their territories (*Gramapurohit, Gosavi & Phuge, 2011*), and we witnessed a single event of aggression where the resident male chased off an intruder (Video S8).

We monitored 15 egg clutches from the moment of deposition until hatching. Of the 15 clutches, 12 were eaten before hatching took place. These cases of predation were rarely observed but could easily be distinguished from hatched clutches, because in hatched

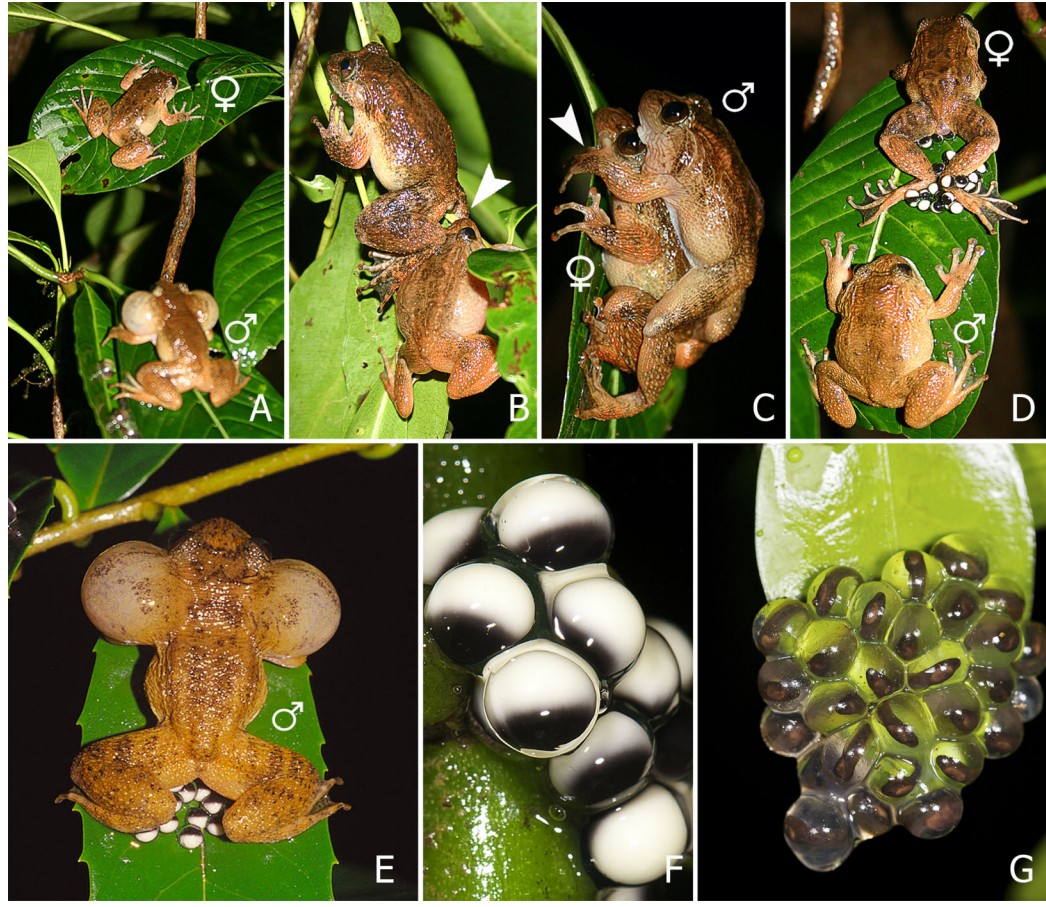

**Figure 2** (A–E) Sequence of breeding behaviour in *Nyctibatrachus humayuni*; (F–G) Egg development. (A) Female approaches a calling male. (B) Female touches male just before the dorsal straddle (arrow indicates the position of female's leg on male's head). (C) Male mounts the female in a dorsal straddle, and most likely sperm is released on her back during this moment (arrow indicates the male's hand positioned on the leaf, but not clasping the female). (D) Female deposits eggs and remains motionless with her hind legs stretched around the eggs. The male is mostly seen sitting close-by without any physical contact with the female. (E) After the female leaves the oviposition site, the male sits on or near the eggs and continues to call. (F) Freshly laid eggs, pigmented (egg diameter 3.5 ± 0.2 mm, n = 20). (G) Developing embryos on the 19th day, just before hatching out of the eggs.

clutches, the jelly remained visible on the substrate after hatching, in contrast to instances of predation, where most of the jelly was removed. In one of the remaining three clutches, hatching started 18 days after oviposition and all larvae had fallen into the stream by the 19th day. In the other two clutches, hatching started after 19 days and finished one and two days later (Figs. 2F and 2G).

## Male advertisement calls

Male advertisement calls are composed of two distinct parts, with the first part being non-pulsatile and the second part having a pulsatile temporal structure (Fig. 3A; Audios S1 and S2). A detailed overview of analysed properties, including the variation both within and between individuals, is given in Table S2.

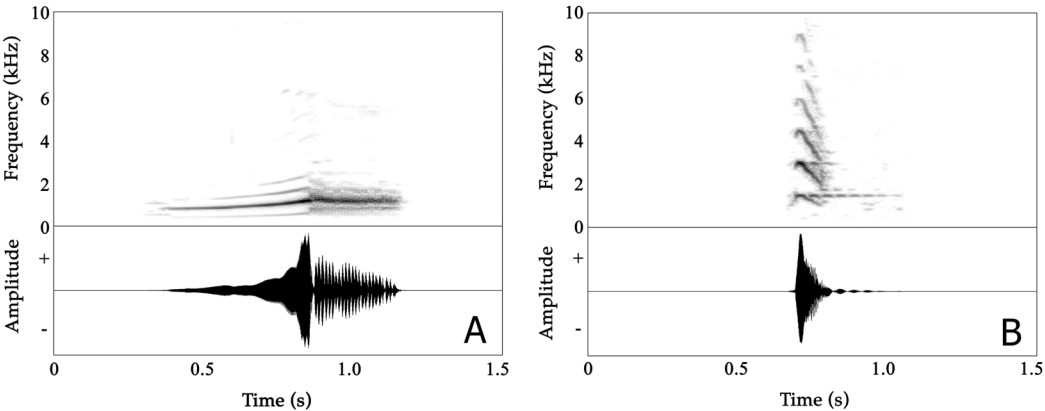

**Figure 3 Spectrograms (above) and oscillogram (below) of *Nyctibatrachus humayuni* calls.** (A) Male advertisement call. (B) Female call.

The mean call duration was 532 ms (range: 483–630 ms). On average, calls reached their full amplitude in 268 ms (rise time) and decreased in amplitude over the last 262 ms of the call (fall time). Both rise time and fall time were approximately 50% of the total call duration (Table S2A). The second part of calls consisted of 16–24 pulses, with a mean pulse period of 13.2 ms and a mean pulse duration of 12.8 ms. These pulses were produced at rates of 74–84 pulses/s. Individual pulses had mean rise and fall times of 4.7 and 8.1 ms, respectively (Table S2C).

The mean dominant frequency, measured over entire calls, was 1.34 kHz (1.11–1.45 kHz) with relatively low variation between calls produced by the same individual ($CV_w = 3.73\%$) compared to the magnitude of variation observed among the eight individuals ($CV_b = 8.19\%$). The dominant frequency of the first, non-pulsatile part of the call ranged between 0.91–1.45 kHz while in the second, pulsatile part the dominant frequency ranged between 1.17–1.47 kHz (as calculated from first pulse, middle pulse and last pulse) (Table S2). Frequency modulation is present in the first, non-pulsatile part of the call, but absent in the second, pulsatile part. The dominant frequency increased by approximately 30% during the first part of the call, from 1027 Hz (mean dominant frequency 1) to 1336 Hz (mean dominant frequency 4) in less than 300 ms (Table S2B).

Results of correlation tests are presented in Table S3. SVL and body mass of the different recorded males and corresponding temperature data is presented in Table S4. The mean (± SD) dry bulb and wet bulb air temperatures during our study were 22.2 ± 0.6 °C and 22.7 ± 0.4 °C, respectively. There were no correlations between these temperatures and the acoustic properties analysed. Though temporal properties are frequently correlated with temperature, we attribute this lack of temperature effects on call properties to the very small temperature variation across our recordings (< 2.1 °C dry bulb air temperature; < 1.1 °C wet bulb air temperature).

Several spectral properties were correlated with our measures of body size. The overall dominant frequency of the entire call was significantly negatively correlated with body mass (Table S3A). The dominant frequencies measured separately over the first and second halves of the call were also significantly negatively correlated to body mass

(Tables S3B and S3C). The overall dominant frequency of the entire call was also negatively correlated with the SVL (Table S3A). The overall dominant frequency, dominant frequency 2, and dominant frequency 4 of the 1$^{st}$ half were significantly negatively correlated with SVL (Table S3B). The overall dominant frequency and dominant frequency of the first and last pulses of the second, pulsatile part of the call were also significantly negatively correlated with SVL (Table S3C). The correlation of dominant frequency 1 and dominant frequency 3 with SVL were marginally non-significant in the first half of the call (Table S3B). Only one temporal property, pulse 50% rise time of maximum amplitude pulse, was correlated with body mass (Table S3C).

With only one exception, there were no significant correlations between body condition and the acoustic properties measured in this study (Table S3). The exception was the temporal property of fall time of the first part of the call, which was significantly negatively correlated with body condition (Table S3B). Several spectral properties had reasonably strong negative correlations with body condition ($-0.69 \leq r \leq -0.62$), but these correlations were not quite significant ($0.06 \leq P \leq 0.10$; Table S3B). Body condition can sometimes be strongly related to certain temporal call properties associated with higher energetic demands, such as fast pulse rates (*Jakob, Marshall & Uetz, 1996*). Although overall pulse rate was not related to body condition (Table S3C), two measures of individual pulse period (= 1/pulse rate) had correlations with body mass that approached significance ($r = -0.69$, $P = 0.06$; Table S3C). We attribute the general lack of significant effects of body condition on several analysed call properties to our small sample size.

## Female call

Female *Nyctibatrachus humayuni* calls were only rarely observed (four individuals), as females do not seem to vocalize on a regular basis. A single female calling from a height of 2.5 m above the ground was recorded. It produced about 50 calls in about 30 minutes. These calls sounded similar to the calls we heard other females produce (Table S5; Video S9). The female call is quite distinct from the male advertisement call (Fig. 3B; Audio S3; Video S9). The call is short, consisting of a single note, with a mean call duration of 83 ms and a rapid onset (call rise time = 16 ms). The call's frequency spectrum is characterized by several frequency peaks, with the lowest three peaks having mean frequencies of 1.45 kHz (Dominant frequency 1), 2.90 kHz (Dominant frequency 2) and 4.37 kHz (Dominant frequency 3), respectively. The overall dominant frequency of the call was 2.85 kHz (range 1.39–3.10 kHz). Based on measures of within-individual CVs, spectral properties were less variable ($2.53 \leq CV_w \leq 12.37\%$) compared to temporal properties ($18.30 \leq CV_w \leq 58.04\%$). The call 50% rise time and call 50% fall time were more variable compared to other temporal properties. During two of our observations of female calling behaviour in *N. humayuni*, the female only initiated calling when she had not succeeded in reaching a calling male after trying for considerable time (more than 30 minutes). On one of these occasions the male immediately changed his position as soon as the female called. Although the male's response resulted in him calling from a new position closer to the stream and to the female, she still failed to locate him.

## DISCUSSION

### New form of amplexus

The loose amplexus observed here in *Nyctibatrachus humayuni* differs from all previously described amplexus types in anurans (*Duellman & Trueb, 1986*: 68–70) (Figs. 4A–4F). Our observations indicate there is some similarity between amplexus in *N. humayuni* and that observed in some mantellid frogs, whereby the male sits with his abdomen on the female's head (*Blommers-Schlösser, 1975*; *Glaw & Vences, 2007*: 144, 186, 200, 204; *Altig, 2008*). *Duellman & Trueb (1986*: 69) defined this type of amplexus as a straddle, and it was later more specifically called a head straddle (*Savage, 2002*: 166). The form of contact described in this study resembles this head straddle, but the male is positioned lower on the female, with his abdomen placed on her lower dorsum. When mounted, the male rests on the female without grabbing her tightly, but instead uses his hands to hold on to a leaf, branch or tree trunk (Figs. 1C, 2C and 4J–4K). On some occasions his hands may rest on her arms or hands (Figs. 4H–4I). *Gramapurohit, Gosavi & Phuge (2011)* considered this to be cephalic amplexus, similar to that present in some dendrobatids. Because the male does not press the backside of his hands against the female's throat, we believe this term should not be used (*Wells, 1980*). We therefore propose to name this behaviour a *dorsal straddle*. A dorsal straddle can be defined as a loose form of amplexus in which the male sits on top of the female with his abdomen positioned on her lower dorsum. The male does not, however, grasp the female under her armpits or head, but instead places his hands on the leaf, branch or tree trunk the pair is sitting on (Figs. 1C and 2C). At the moment, dorsal straddle is known to occur in *N. humayuni* only, but observations made in other *Nyctibatrachus* species (*Biju et al., 2011*) might also correspond with this newly-defined type of amplexus.

We speculate that a loose form of contact, with the male holding on to the substrate rather than to the female, as seen during a dorsal straddle, might function to avoid falling and interrupting mating. In *Guibemantis depressiceps* (*Boulenger, 1882*), which uses a head straddle during amplexus, pulsing movements in the male flanks were observed during a short period of physical contact (*Blommers-Schlösser, 1975*), similar to the ones observed here for *Nyctibatrachus humayuni* (Video S4). The function of these pulsations is unclear, but similar movements during amplexus have been observed in several other frog species (*Savage, 1932*; *Rabb & Rabb, 1963*; *Weygoldt, 1976*; *Cui et al., 2010*). It has been proposed that these movements might stimulate ovulation by the female (*Duellman & Trueb, 1986*: 71; *Cui et al., 2010*).

### Moment of fertilization

The loose contact during amplexus, and the lack of any physical contact between both sexes during oviposition, that we have described is rarely seen among anurans. Our observations significantly extend those made by *Gramapurohit, Gosavi & Phuge (2011)* of the same species. *Gramapurohit, Gosavi & Phuge (2011)* observed a subsequent step after oviposition, in which the male placed himself on top of the eggs, and they considered this to be the moment of fertilization (*Gramapurohit, Gosavi & Phuge, 2011*). In our study,

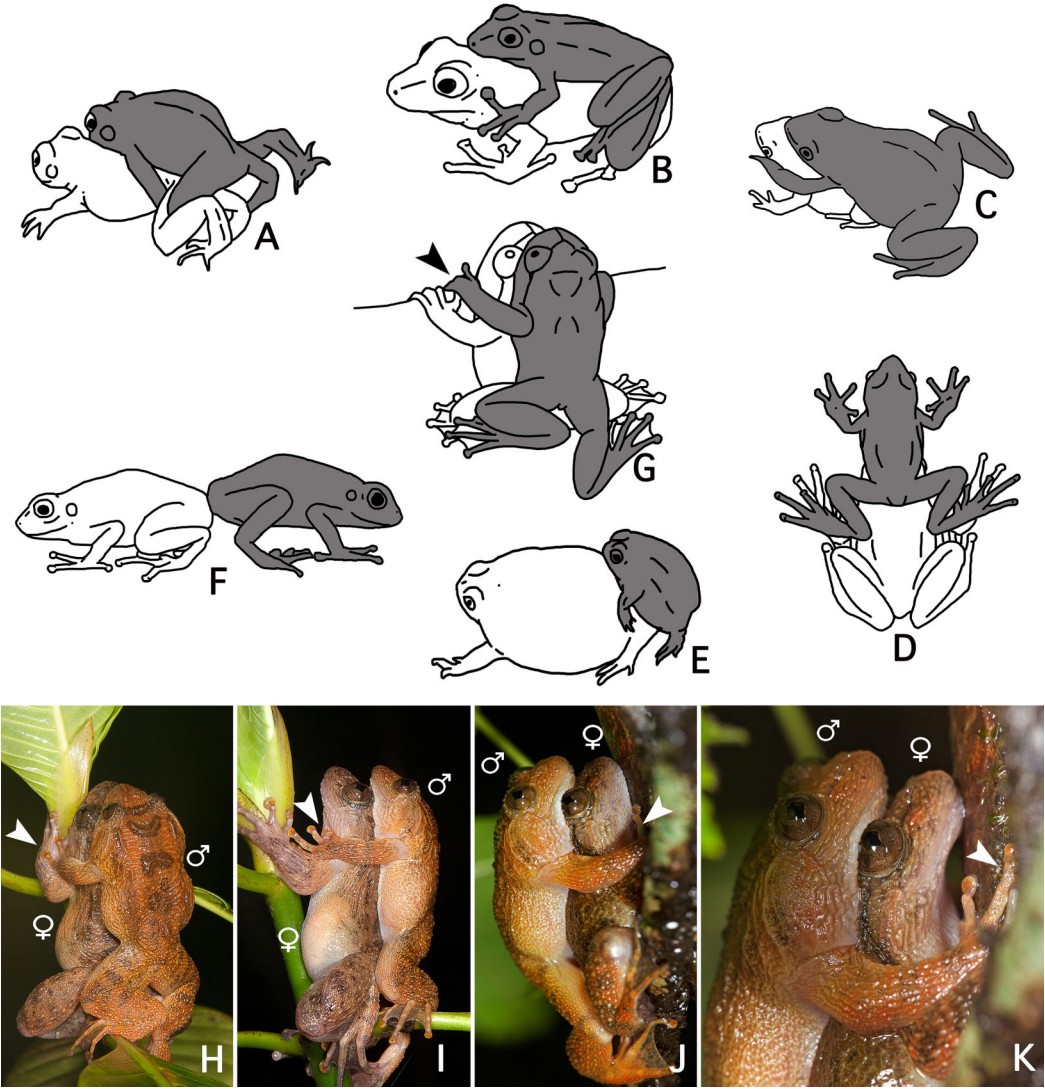

**Figure 4 A comparison of known amplexus positions found in anuran amphibians with the new amplexus mode in *Nyctibatrachus humayuni*.** (A–F) Known amplexus positions. (A) Inguinal. (B) Axillary. (C) Cephalic. (D) Head straddle. (E) Glued. (F) Independent (adapted from *Duellman & Trueb, 1986*: 69). (G–I) Dorsal straddle, with the male's hands on sides of the female's head but not clasping the female. (G–H) Dorsal views. (I) Side view. (J–K) Dorsal straddle, with the male's hands holding a twig but not clasping the female. (J) Side view. (K) Close-up of the side view. Arrows indicate the male grasping position. The male is drawn grey and the female white.

however, all clutches covered with a Ziploc® bag immediately after oviposition showed a 100% fertilization rate, proving that semen release must happen before clutch deposition, since the male dismounts the female just before oviposition (Videos S5 and S7). This is remarkable, since amplexus is considered to increase chances of successful fertilization by synchronizing egg deposition and semen release (*Duellman & Trueb, 1986*: 68–70).

A complete lack of physical contact between male and female during egg deposition is, therefore, rarely seen in anurans (*Wells, 2007*: 458, 513). Our findings are inconsistent with the idea of the male fertilizing the eggs after deposition, as postulated by

*Gramapurohit, Gosavi & Phuge (2011)*, and instead suggest that semen release must take place during the dorsal straddle. We hypothesize that the male releases his sperm on the female's back and the eggs are subsequently fertilized by the semen running down her back and hind legs. This hypothesis is supported by the observation that female remains motionless after egg deposition, for 479 s on average, with her hind legs stretched around the freshly deposited clutch (Figs. 1E and 2D; Video S6). Furthermore, similar behaviour has been reported for *Blommersia wittei* (*Guibé, 1974*), *B. blommersae* (*Guibé, 1975*), *Guibemantis depressiceps* and *G. liber* (*Peracca, 1893*), among others (*Blommers-Schlösser, 1975*; *Glaw & Vences, 2007*: 144, 186, 200, 204; *Altig, 2008*). In these Malagasy frogs, males engage in a head straddle with females and dismount the female before she finishes egg deposition.

In *Guibemantis depressiceps*, multiple males were observed sitting on a female's head during a single egg deposition event, and the female exhibited lethargic behaviour, remaining stretched around the clutch for at least an hour after deposition had finished (*Altig, 2008*). For these mantellid frogs, the hypothesis of sperm running from the female back has also been assumed (*Blommers-Schlösser, 1975*; *Altig, 2008*). The exact moment of semen release could not be observed, as this was not possible with the naked eye during extremely wet field conditions, a problem also mentioned by *Altig (2008)*. The male might release his semen just before dismounting the female. The eggs are then subsequently fertilized by sperm running down her back. Another possibility is that sperm is gradually released during amplexus and that the eggs are eventually deposited on top of the sperm. This last option has been observed in some species of Dendrobatoidea (*Weygoldt, 1980*). A few anurans make use of internal fertilization (*Townsend et al., 1981*; *Stephenson & Verrell, 2003*), but for *N. humayuni* this is very unlikely as males lack an intromittent organ and only limited contact between the two sexes exists.

In *Nyctibatrachus petraeus* there is no physical contact at all, and it is believed that the male fertilizes the eggs after the female has deposited the clutch (*Kunte, 2004*). In this species the female is reported to immediately leave after egg deposition (*Kunte, 2004*), contrasting with our findings of *N. humayuni* females. *Gururaja et al. (2014)* do not mention the moment of fertilization in *N. kumbara*, but it is unlikely to happen as in one of the methods described above. Since the female deposits the eggs from an upside-down position, sperm cannot run down from her back onto the eggs. Furthermore, the male only touches the eggs with his hands after deposition, to cover them with mud (*Gururaja et al., 2014*). Future studies regarding the reproductive behaviour of *Nyctibatrachus* should aim to pinpoint when semen is released and how fertilization is achieved.

## Egg development and egg attendance

Of the egg clutches monitored, 80% were eaten by predators before hatching. During a single occasion, a *Boiga* sp. (tree snake) was observed eating a *N. humayuni* egg clutch (Video S10). Another water snake, *Rhabdops olivaceus* (*Beddome, 1863*), was frequently observed around the oviposition sites but no direct observation of predation was made. *Gramapurohit, Gosavi & Phuge (2011)* reported an egg stage duration of 11–13 days in their study population, which is much shorter than our observation of 18–21 days

(Figs. 2F–2G). An egg stage of 12–15 days and 8 days were observed for *Nyctibatrachus petraeus* and *N. kumbara*, respectively (*Kunte, 2004*; *Gururaja et al., 2014*), while for *N. aliciae* an egg stage of 8–9 days was observed in a clutch studied in captivity (*Biju et al., 2011*). As egg development duration is negatively correlated with temperature within a certain optimal range (*Duellman & Trueb, 1986*: 120–124; *Wells, 2007*: 124, 499), a higher number of observations and associated temperature data will be needed to address this variation.

Males were seen each night in the same location, sitting near or on the developing clutches that had resulted from earlier successful mating encounters. Egg attendance by the male, or both male and female, is known for all *Nyctibatrachus* species in which reproductive behaviour has been studied (*Kunte, 2004*; *Biju et al., 2011*; *Gramapurohit, Gosavi & Phuge, 2011*; *Gururaja et al., 2014*). The function of egg attendance has not yet been studied in this genus, but it might prevent desiccation or reduce predation risk by arthropods, both of which have already been confirmed in other anuran taxa (*Crump, 2015*). More specific parental care has been documented for *N. kumbara* and *N. grandis*. *Nyctibatrachus kumbara* males cover the egg clutch with mud, possibly preventing dehydration or providing camouflage against predation (*Gururaja et al., 2014*). Males of *N. grandis* have been observed inflating their body upon disturbance and making attempts to bite would-be predators, including human observers (*Biju et al., 2011*).

## Male call

The advertisement calls of only a few *Nyctibatrachus* species have been described (*Kuramoto & Joshy, 2001*; *Gururaja et al., 2014*). *Gururaja et al. (2014)* provide a brief description of the calls of *N. kumbara*, *N. jog* and *N. kempholeyensis* (*Rao, 1937*), while a concise description of the call of *N. major Boulenger, 1882* is given by *Kuramoto & Joshy (2001)*. In *N. kumbara* and *N. kempholeyensis* two distinct call types are distinguished; in *N. kumbara* one of these types was produced more frequently when a female was present (*Gururaja et al., 2014*). Our recordings were made in the absence of females. Therefore, we cannot exclude the possibility that in *N. humayuni* too, distinct call types are present. The advertisement call from male *N. humayuni* described here is complex, consisting of an initial, unpulsed part with frequency modulation and a second, pulsed part without frequency modulation (Fig. 3A; Table S2). The call of *N. jog* also has a pulsed second part of the call (*Gururaja et al., 2014*). The call of *N. humayuni* can be distinguished from the call of *N. major*, the type I call of *N. kumbara*, and both types of call of *N. kempholeyensis* by call duration. With an average duration of 0.53 s, the call is longer than the type I call of *N. kumbara* (0.11 s) and the call of *N. major* (0.05 s), while being substantially shorter than both call types of *N. kempholeyensis* (5.17 and 11.69 s for the type I and type II calls, respectively). Although call duration is similar between *N. humayuni* and the call of *N. jog* and the type II call of *N. kumbara*, the dominant frequency of *N. humayuni* calls is lower (1.33 kHz) than that of *N. kumbara* type II calls (1.53 kHz) and *N. jog* calls (1.51 kHz).

Our results show a significant negative correlation between male body size and dominant frequency (Table S3). This is the case for many anuran species, in which

relatively larger individuals' calls are characterized by relatively lower dominant frequencies. Larger males tend to have more massive vocal cords and, consequently, produce lower-frequency calls (*Martin, 1971*). In some species, females show a preference for low frequency calls and hence larger males (*Ryan, 1980*; *Wollerman, 1998*). However, experimental studies have also shown that in other species no preference is given to lower frequencies (*Rosso, Castellano & Giacoma, 2006*). Furthermore, female preference can vary between different populations within a single species (*Schrode et al., 2012*). Within *Nyctibatrachus,* only a few species' calls have been described, and no data on female preferences for spectral properties are available, making it impossible to draw conclusions without further bioacoustic research on this group.

## Female call

The discovery of a female call is remarkable, as this is rarely observed in frogs and toads. Female calling behaviour has so far been reported in–to our knowledge–only 25 anurans, representing less than 0.5% of the total of 6583 anuran species that are currently recognized (25/01/2016) (*Frost, 2015*). An overview of the species for which a female call has been reported is given in Table S6. Female calling in *Nyctibatrachus humayuni* was observed only briefly and on just four occasions over a total of 40 nights in the field, compared to the almost permanent presence of male advertisement calls. The female call of *N. humayuni* is shorter and less complex than that of the male, consisting of a single, unpulsed note (Fig. 3B; Video S9). A short and less intense female call is also observed in other anuran species (*Given, 1987*; *Emerson, 1992*; *Cui et al., 2010*) and can, in part, be explained by the smaller size of laryngeal and oblique muscles of the female (*Emerson & Boyd, 1999*). A female call is not known from any other species of *Nyctibatrachus.* Since documentation of the presence of female calls can be done with more confidence than their absence (*Wells, 2007*: 282), female vocalizations might be more common than currently believed, both among *Nyctibatrachus* species and across all anuran taxa more generally (*Emerson & Boyd, 1999*).

Observed male responses to female calls include positive phonotaxis and changes in male vocalization rate (*Emerson & Boyd, 1999*; *Shen et al., 2008*; *Cui et al., 2010*; *Wang et al., 2010*), suggesting that female vocalization is important in mate location and recognition. In this and other *Nyctibatrachus* species, calling males are often located in difficult to reach locations (e.g. vegetation overhanging the water) that might be accessed only by leaping towards them directly from the stream. It is likely that in such a situation the female will mainly rely on acoustic cues to localize the calling male. Intensified male calling, or the male shifting to another position, in response to a female call might then prove beneficial in successfully locating a mate. This hypothesis is only speculative at present, and additional observational and experimental studies will be required to test it. Several other potential functions have been attributed to female calling behaviour, such as signalling receptivity (*Tobias, Viswanathan & Kelley, 1998*; *Shen et al., 2008*), distinguishing satellite males from territorial ones (*Given, 1993*) and inciting male-male (*Judge, Swanson & Brooks, 2000*) and female-female competition (*Bush, 1997*; *Bosch, 2002*). Few of these hypotheses, however, have been tested experimentally (*Wells &*

*Schwartz, 2007*). Both positive and negative phonotaxis has been observed in male *Xenopus laevis* (*Daudin, 1802*) in response to two distinct female call types (*Wang et al., 2010*). Distinct female call types have also been described for *Alytes cisternasii* (*Márquez & Verrell, 1991*). Together, the scarce data available on female vocal behaviour and the variability observed in male responses suggest a complex of multiple, non-exclusive functions rather than a sole explanation.

### Diversity in reproductive behaviours

Our results, combined with other recent behavioural studies (*Kunte, 2004*; *Biju et al., 2011*; *Gururaja et al., 2014*), show a striking interspecific diversity in reproductive behaviours within *Nyctibatrachus*. As the reproduction of many species in this genus has not yet been studied, the extent of variation is likely to be even greater. Our limited understanding of this diversity is evidenced by the high number of recent publications reporting new modes of breeding and associated reproductive behaviours (*Zhang et al., 2012*; *Gururaja et al., 2014*; *Iskandar, Evans & McGuire, 2014*; *Crump, 2015*; *Seshadri, Gururaja & Bickford, 2015*; *Senevirathne et al., 2016*). Within *Nyctibatrachus*, variation in the types of amplexus is especially remarkable, ranging from a normal axillary amplexus to a complete lack of physical contact between the sexes (*Kunte, 2004*; *Gururaja et al., 2014*). The use of amplexus is the ancestral state in anuran amphibians and enables juxtaposition of male and female cloaca to ensure successful fertilization (*Duellman & Trueb, 1986*: 68–70; *Wells, 2007*: 452–458). Secondary loss of amplexus was already shown for a limited number of anuran taxa (*Limerick, 1980*; *Brown et al., 2008*; *Zhang et al., 2012*) and a short, loose contact (head straddle) similar to the dorsal straddle reported here for *N. humayuni*, was also found in several *Mantidactylus* species (*Blommers-Schlösser, 1975*; *Altig, 2008*). Many males of both *Mantidactylus* and *Nyctibatrachus* possess distinct femoral glands (Fig. S2B) (*Vences et al., 2007*; *Biju et al., 2011*). The function of these structures in *Nyctibatrachus* has not yet been addressed, but recently it was shown that the femoral glands of *Mantidactylus* secrete volatile pheromones (*Poth et al., 2012*; *Hötling et al., 2014*). Further studies are required to discover and understand the full extent of variation and the evolutionary advantage of these highly diversified behaviours within *Nyctibatrachus*, especially regarding the modification and loss of amplexus.

## CONCLUSION

The breeding behaviour of *Nyctibatrachus humayuni* has several unique elements: a new type of amplexus, the release of semen before oviposition and the presence of a female call. These findings further highlight the tremendous variation present in the reproductive behaviour of anuran amphibians. *Nyctibatrachus* frogs are one of several unique taxa in the Western Ghats biodiversity hotspot, which is heavily threatened by anthropogenic activities (*Myers et al., 2000*; *Bossuyt et al., 2004*; *Van Bocxlaer et al., 2012*). A good understanding of each species' ecology, including reproduction, is of major importance for planning and successfully implementing conservation strategies. Additional studies further exploring the unique and diverse behaviour in *Nyctibatrachus* frogs are, therefore, badly needed. Special attention should be paid to describing the amplexus type,

determining the moment of fertilization and assessing the presence and function of female calling behaviour.

## ACKNOWLEDGEMENTS

We would like to thank Franky Bossuyt for valuable inputs during the initial phases of this study; Kim Roelants for the basic illustrations used in Fig. 1; our guide, Shanker Mama, for assistance in the field; M. M. Anees and V. Prasad for lab support; Ramit Verma for help in compiling the videos; three anonymous reviewers and the academic editor for their comments and suggestions to improve this manuscript. We are grateful to the Maharashtra State Forest department for providing permits to conduct this study.

### Funding

This study was partially supported by the following grants to SDB: CEPF 'Project 55918/2009,' USA; University of Delhi Research and Development Grants '2011/423' and '2015/9677'; DST Purse Phase II Grant 2015, Department of Science and Technology, Ministry of Science and Technology, Government of India. BW was funded by Prof. Franky Bossuyt, Amphibian Evolution Lab (Vrije Universiteit Brussel). MAB was funded by a Fulbright-Nehru Award. The funders had no role in study design, data collection and analysis, decision to publish, or preparation of the manuscript.

### Grant Disclosures

The following grant information was disclosed by the authors:
SDB: CEPF, USA: Project 55918/2009.
University of Delhi Research and Development: 2011/423 and 2015/9677.
DST Purse Phase II 2015.

### Competing Interests

The authors declare that they have no competing interests.

### Author Contributions

- Bert Willaert conceived and designed the experiments, performed the experiments, analyzed the data, contributed reagents/materials/analysis tools, wrote the paper, prepared figures and/or tables, reviewed drafts of the paper, participated in fieldwork, prepared videos.
- Robin Suyesh performed the experiments, analyzed the data, contributed reagents/materials/analysis tools, wrote the paper, prepared figures and/or tables, reviewed drafts of the paper, participated in fieldwork, prepared videos.
- Sonali Garg performed the experiments, analyzed the data, wrote the paper, prepared figures and/or tables, reviewed drafts of the paper, participated in fieldwork, Prepared Videos.
- Varad B. Giri performed the experiments, contributed reagents/materials/analysis tools, reviewed drafts of the paper, participated in fieldwork.

**PeerJ** _______________

- Mark A. Bee analyzed the data, wrote the paper, prepared figures and/or tables, reviewed drafts of the paper.
- S.D. Biju conceived and designed the experiments, performed the experiments, analyzed the data, contributed reagents/materials/analysis tools, wrote the paper, prepared figures and/or tables, reviewed drafts of the paper, participated in fieldwork, prepared videos.

### Animal Ethics

The following information was supplied relating to ethical approvals (i.e., approving body and any reference numbers):

This study was conducted with permissions and guidelines from the responsible authorities in the State Forest Department of Maharashtra. Study permit: D-22 (8)/Research/4543/2012-13, dated 1-03-2012. This study did not sample animals for any captive or laboratory studies. All observations were made in the wild. Recorded frogs were released back at their calling site immediately after measuring the body size and body mass.

### Data Deposition

The raw data is contained in the body of the manuscript and Supplemental Information.

### Supplemental Information

Supplemental information for this article can be found online at http://dx.doi.org/10.7717/peerj.2117#supplemental-information.

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
