# Peer review of "A unique mating strategy without physical contact during fertilization in Bombay Night Frogs (Nyctibatrachus humayuni) with the description of a new form of amplexus and female call"

_PeerJ, doi:10.7717/peerj.2117_

## Round 0.1 · original submission · Major Revisions

Please revise your manuscript, paying close attention to all of the reviewers' comments.

Reviewer 1 ·

Basic reporting

No comments

Experimental design

No comments

Validity of the findings

No comments

Additional comments

This is an interesting paper that reports a remarkable new form of mating behavior in frogs, as well as a new example of females producing calls. I have only a few minor comments:

Table 1 seems like overkill to me; having such a long table to define terms interrupts the flow of the text and at least should be moved to the supplementary material.

Line 97: Taxonomy is out of date. L. blythii studied by Emerson in Borneo is now L. leporinus (same for Table 2). At least some of the so-called voiceless frogs do in fact have male advertisement calls.

Line 298: The female produced about 50 calls over what time period? The authors do not include calling rate as a variable anywhere in the paper, which is something readers might want to compare with other species.

Line 410: I hate the term “predated,” which means to come before in time, as in: Greek civilization predated Roman civilization. Why not use regular English and say “were eaten by predators”?

Line 516: Diversity of what? Need a more explanatory heading.

Lines 810-812: The two Wells citations are out of chronological order because of an extra space after the author’s initials in Wells 1980.

The authors missed a very interesting paper on female calls, including very precise phonotaxis by males toward female calls: Shen et al. 2008. Ultrasonic frogs show hyperacute phonotaxis to female courtship calls. Nature 453:914-917.

Reviewer 2 ·

Basic reporting

Although generally well written, I consider the introduction and the discussion overly long - just because online-only journals do not impose restrictions on manuscript length, the authors should not abuse this freedom and still stick to certain "values" from conventional journals, such as concise wording and phrasing and refrain from a narrative style. I assume that a professional copy editor/proof reader could easily shorten the text by 1-2 manuscript pages (so ~0.5-1 typeset pages).

The authors also should stay more with the title of their manuscript, and not meander off into a broad review on nyctibatrachid behaviour or anuran female calling - these would be proper manuscripts on their own, but should not be sneaked into a study that aims (at least in its title) at reporting very specific phenomena in one single species. To put the actual findings of this study into a broader context of course is perfectly fine and wanted, but this shout happen within bounds.

On the other hand, while touching a multitude of aspects in the introduction and discussion, the authors are rather reluctant in providing references to primary literature - Duellman & Trueb 1986, Gerhard & Huber 2002, Narins et al. 2007 and Wells 2007 are excellent books - however, they are referred to here far too often, when the authors in fact should have looked up (at least, if not read) the primary literature on these topics, which actually usually will be given in these books. Excessive citing of books is fine for grant applications and/or when there are hard limits on the number of references used, but it is not needed nor desirable in original research articles.

The authors should check all instances of given species names and apply a strict strategy when to abbreviate the genus name - convention usually is to only write it in full upon first mention or if there is ambiguity; some journals wish for full species names once in each section of a manuscript - both is not the case here, but full and abbreviated names appear to be used rather randomly. Also PeerJ in its guidelines "encourages" to give taxonomic authorities with species names - the authors should consider doing so.

It is journal style to spell out only numbers 1-9 unless when used with units - the authors should follow this rule.

I am not sure, whether the reporting of all call parameters is beneficial for the readability of the manuscript. Personally I would prefer a presentation, where only the basic/most important call parameters are presented in the main manuscript, while the full body of parameters is put to the supplements. Further, a graphical representation of the measurements in the spectrogram and oscillogramm might by more meaningful to the reader, than the bare verbal descriptions and numbers.

Experimental design

The experimental part of the manuscript is rather short. The sample size of 5 manipulated clutches is very small - however, as the authors show that the clutches were successfully fertilized *despite* their manipulation which prevented direct contact of the male with the clutch post-amplexus, there is no doubt, that fertilization *in these cases* must have happened before! However, due to the small sample it cannot be inferred with statistical certainty, that this is the norm in this species!

Validity of the findings

Related to the experimental design, and in the face of the time spent on this study in the field, I think it is a pity, that the authors did not go the full length to provide the study with a good sample size on the moment/mode of fertilization and female calls. This would have made it easier to recognize their findings as being the general norm for this species and not mere anecdotal observations - I leave it to the editor to decide in the trade-off between the need to report novel observations and the need for larger sample sizes to make sound generalisations.

Additional comments

L39: the proper plural of "amplexus" to be used here is "amplexi" - probably needed also somewhere else in the manuscript

L131: you should mention, which specific aspects you deem to be previously unreported in anurans

L143: I do not understand what "opportunistic searches" should be - do you mean "chance encounters"?

L147: At this point it is hard to understand your rationale for locking males from their clutches - please explain here, at least briefly your motivation!

L162ff: You use very loose wording to describe the spatial setting of the focal individuals - "widely spaced", "large", "exactly the same place" - please be more specific! Further, from my experience with several territorial, hyper-dispersed frog species, assumed site fidelity can be a very poor proxy for individual identity - can you provide or cite more information, why you are sure that you observed/recorded distinct individuals?!

L222ff: Again very loose wording - "In at least half...", "usually", high", "close vicinity", "different position", "usual spot" - please be more specific!

The last two comments apply to several instances throughout the manuscript - please check carefully!

There are several inconsistencies between the numbers reported in the table, and the numbers given in the text - e.g.: male call, overall dominant frequency: 1.34 vs 1.33 kHz; part 2 1.17-1.53 vs. 1.17-1.47 kHz - there might be other instances, please check carefully all values in the tables and text!

L276: Are you reporting standard deviations as your measure of spread? Please specify!

L359: I do not see why this should *logically* (sensu stricto) follow form the observation - maybe it was apparent, or obvious, or seemed natural.

L427: Only in - awkward wording, please rephrase

L440: See before - it is a pity with such an extensive study, and the authors even hypothesizing about the possibility of a close encounter call (or other calls), that they did not take the time to go the full mile...

Table 4 - see my initial remark on "review vs. specific report" - I dont see how this table brings forward the manuscript. Also, is the femal call of D. microcephala really only reported in Wells 2007?

Can you give any information on sexual dimorphisms in your study species? How did you determine the sex of your calling (alleged) females?

Several instances of "interesting" appear rather narrative and inappropriate for a formal scientific text.

The funding organisations are mentioned in the acknowledgements as well as in the appropriate section!

L810: Wrong order for Wells 2007 & Wells 1980

Reviewer 3 ·

Basic reporting

The manuscript is well written in general, appropriate references are cited, and enough background information is given to justify the study. I have made some minor comments in both form and content in order to improve certain passages. The figures illustrate clearly what the authors are arguing. Altogether this is a very nice piece on natural history of an understudied taxon and, as such, I very much value the information that the authors are providing. Natural-history works tend not to be given much priority nowadays, but I am glad to see that there are still scientists willing to explore this very relevant aspect of scientific research, as it is vital for the generation of new hypothesis and ideas.

Experimental design

The observations, recordings and experiment with clutches were done appropriately

Validity of the findings

The call properties of the female call are based on one individual only, so they must be taken cautiously. However, the authors appropriately address this limitation. The rest of the findings are well discussed and placed within the relevant previous knowledge on the topic

Additional comments

Altogether this is a very nice piece on natural history of an understudied taxon and, as such, I very much value the information that the authors are providing. Natural-history works tend not to be given much priority nowadays, but I am glad to see that there are still scientists willing to explore this very relevant aspect of scientific research, as it is vital for the generation of new hypothesis and ideas.

Find below some minor comments that i believe would contribute to improve the manuscript:

L37: Add an 's' to the word behaviour

L39: change ‘a variety of’ to ‘several’

L44: Can this really be called amplexus if there is no clasping? I would suggest you reword this phrase and call it –as the title says- ‘mating strategy’

L130: Add an 's' to the word behaviour

L139: replace ‘to study’ with ‘studying’

L164: “males of this species are territorial” — could you provide a reference to back this up?

L219: change ‘have’ to ‘had’

L273: In this section I would suggest the authors to explore correlations between call properties and the body condition index (Jakob, Marshall, & Uetz, 1996). Given that SVL and mass tend to be positively correlated, it results interesting –an in some respect more appropriate- to evaluate the relationship between call parameters and that mass that is not explained by SVL, but tells whether the frog has bigger reserves. This is particularly relevant for temporal properties of the call, which involve a presumably higher energetic demand such as pulse repetition rate.
Jakob, E. M., Marshall, S. D., & Uetz, G. W. (1996). Estimating fitness: a comparison of body condition indices. Oikos, 77, 61-67.

L298: I am not comfortable with the fact that you name the female’s call ‘advertisement call’, especially with such limited sample size. I would suggest you simply call it ‘female call’ and explain in the discussion section why you believe this call would have an ‘advertisement’ function.

Line 313: replace ‘in locating him’ with ‘to locate him’

Line 356: this needs to be reworded; no physical contact between the sexes exists during oviposition for dendrobatid frogs either, for example.

Line 366: in this line you say that it is only rare rather than unique. This needs to be unified for consistency (it is either unique or rare)

Line 390: be consistent with terminology! You said above this was a straddle rather than an amplexus…

Line 468: I agree, and this is a fantastic observation!

Line 491: “Observed male responses to female calls include positive phonotaxis and changes in male vocalization rate” —> could this be a courtship call rather than an advertisement call? I agree with the segment of your discussion section where you acknowledge that this call could have several functions rather than one, and for that reason I suggest you call it simply ‘female call’ instead of ‘female advertisement call’

Line 516: Rather than only ‘Diversity’ I suggest this subheading to be ‘Diversity in reproductive behaviours’

You should also cite Crump's classic paper on anuran reproductive modes:
Crump, M. L. (1974). Univ. Kansas Mus. Nat. Hist. Misc. Publ., 1-65.

---

## Round 0.2 · Minor Revisions

Thank you for your revisions. At this point, only minor changes are necessary, as outlined in the review.

Reviewer 2 ·

Basic reporting

From the extensive list of changes and detailed comments to the reviewers suggestions one can see, that the authors took great care in revising their manuscript, which is basically ready for publication, besides a view very minor suggestions!

I did not object against the general use of books as references, but rather to referring books for very specific information, that is unlikely to be original research from a books author or editor, but rather of a study reviewed in the book. E.g. line 85. However, after rechecking I agree that Wells 2007 indeed is a suitable reference here, also as there is no proper review paper on the amplexus, as you stated. To help the reader to find the relevant passage I suggest to give chapter (here: 10) or even page numbers (here: 456, or 452-458) after the in-line references, as one is somehow lost when trying to find the referred information in these more than comprehensive books, unless being able to automatically search in an electronic version.

I particularly like the authors move to not only include the taxonomic authorities with species names, but also to cite the respective works – only like this taxonomists can receive their due credit!

Experimental design

-

Validity of the findings

-

Additional comments

L39, 77, 85 (probably elsewhere) - No, you did NOT change the plural of "amplexus" to "amplexi", nowhere in your manuscript!

L95 – unclear reflexiveness; “…important to record and describe calls, as they…”

L100 – “with males”

L 203, 205, 238 – For better flow of reading, maker brands and types should be stated in a uniform manner – rather in parenthesis after a general description of the device, like in L 202 for the “digital recorder”

L563 – “…that the female remains…”
Please enter the information on the sexual dimorphism and sex determination of study individuals, as you stated in the rebuttal letter, directly into the manuscript.

---

## Round 0.3 · accepted · Accept

Thank you for your revisions.